# *Lactobacillus salivarius* CML352 Isolated from Chinese Local Breed Chicken Modulates the Gut Microbiota and Improves Intestinal Health and Egg Quality in Late-Phase Laying Hens

**DOI:** 10.3390/microorganisms10040726

**Published:** 2022-03-28

**Authors:** Chang Xu, Fuxiao Wei, Xinyue Yang, Yuqing Feng, Dan Liu, Yongfei Hu

**Affiliations:** State Key Laboratory of Animal Nutrition, College of Animal Science and Technology, China Agricultural University, Beijing 100193, China; xuchang_001@163.com (C.X.); fuxiaowei1005@163.com (F.W.); xinyueyang1124@163.com (X.Y.); fengyuqing2012@gmail.com (Y.F.); liud@cau.edu.cn (D.L.)

**Keywords:** *Lactobacillus salivarius*, laying hens, gut microbiota, egg quality, intestinal health

## Abstract

*Lactobacillus* strains with fine probiotic properties are continuously needed in the laying hen industry to improve the animals’ gut health and production performance. In this study, we isolated 57 *Lactobacillus* strains from the gut microbiota of 17 different chicken breeds in China. We characterized the probiotic features of these isolates, and evaluated the effects of a selected strain, *Lactobacillus salivarius* CML352, on the production performance and gut health of the late-phase laying hens. The results showed that the isolates varied much in probiotic properties, among which *L. salivarius* CML352 displayed high acid and bile salt tolerance, high hydrophobicity, auto-aggregation, and antibacterial activities. Whole genome sequencing analysis showed that CML352 was closely related to a strain isolated from human fecal samples, but had different functional potentials. Dietary supplementary of *L. salivarius* CML352 significantly reduced the Firmicutes to Bacteroidetes ratio, increased the expression of *Muc-2*, and decreased the expression of *MyD88*, *IFN-γ*, and *TLR-4*. Furthermore, strain CML352 reduced the birds’ abdominal fat deposition, and improved egg quality. Taken together, this study indicated that the newly isolated *L. salivarius* strain might be a worthy probiotic with positive impacts on the intestinal health and production performance of late-phase laying hens.

## 1. Introduction

The laying hen industry is one of the key animal husbandries worldwide. However, the late-phase of production (the egg production is below 90%) accounts for a large proportion of the whole period of layer production, during which the production performance and egg quality significantly declined, leading to a restricted economic benefit of layer production [1]. Thus, it is of great significance to improve the performance of laying hens during the late-phase of laying. One pivotal reason for the decreased productive performance of laying hens in the late phase of production could be the corresponding exhaustion of intestinal function [2].

It has been increasingly recognized that intestines play an important role in contributing to the overall health and production performance of layers, and even the poultry industry [3], for it not only allows absorption of nutrients but also acts as a barrier to prevent pathogens and toxins [4]. As a complex structure, a healthy chicken intestinal barrier is made up of four main components: the physical barrier, chemical barrier, immunological barrier, and the microbiological barrier; among them, microbiological barrier is an essential component [5], mainly formed by intestinal bacteria. Evidence has proved that microbiota in chicken intestine can improve production performance by regulating digestion and metabolism [6], and protect the host from pathogen colonization of the gut. A disorder of the gut microbiota promotes the development of diseases such as inflammatory bowel diseases (IBD) [7], metabolic disorders [8], and autoimmune diseases [9]. In recent years, probiotics have been recognized to have beneficial effects, and widely used in poultry husbandry to keep chicken intestinal homeostasis and improve the productivity through altering the composition and functions of gut microbiota in recent years [10,11].

One of the most prominent bacteria used as probiotics belong to the genera, *Lactobacillus* [12]. A wealth of information had been gathered over the past years regarding the beneficial health effects and possible mechanism of actions of *Lactobacillus* on poultry [13]. *Lactobacillus* species could colonize the epithelial tissues in chicken gut [14], and exhibit unique bactericidal activity through the production of organic acids, hydrogen peroxide and bacteriocins [15]. They are also widely recognized for their immunomodulatory effects on both humans and animals [16]. In broilers and layers, the role of *Lactobacillus* strains in affecting the microflora of digestive tract and increasing the growth and production performance of chickens has been revealed [17,18].

Different *Lactobacillus* strains have been proven to have discrepant beneficial effects in chicken, which might depend upon factors such as different bacterial strains, different concentrations, the form of probiotics, the ages of the hens, the appetites of the animals, and the hygiene conditions of the farm [19]. Among them, the strain’s probiotic activity is a very important precondition. Due to natural selection, *Lactobacillus* strains with different origins may evolve different features to adapt their regular living conditions. In recent years, the endogenous *Lactobacillus* from gastrointestinal tract, also known as autochthonous bacteria, is attracting more attention.

Depending on the relationship between the bacteria and the host, the gut microbiota is classified into autochthonous microbiota and allochthonous microbiota. Autochthonous microbiota is relatively safe and generally does not cause a secretion of antibodies in the host, or produces only low levels of antibodies, while allochthonous microbiota can usually trigger a strong immune response [20]. Meanwhile, it is emphasized that the ability of *Lactobacillus* to colonize the intestinal epithelial cells is host-specific [21]. An isolate from the host itself will be more adaptable to support a commensal relationship with the host as compared to bacteria from dissimilar organisms [22]. Thus, it is necessary to isolate and develop autochthonous *Lactobacillus* to be health-promoters for their original host species [23]. Previous studies have confirmed the positive effects of autochthonous *Lactobacillus* strains on animal growth performance, immunity, disease resistance, and gut microbiota balance [24,25].

Based on these findings, we hypothesized that healthy chickens’ digestive systems could be a potential source of *Lactobacillus* strains with fine probiotic properties and a resulting beneficial effect on laying hens. Therefore, in this study, we isolated and evaluated the probiotic properties of autochthonous *Lactobacillus* strains from Chinese local breed chickens. Then, we performed animal experiments to investigate the role of the selected strain in modulating gut microbiota, intestinal health, and egg quality in late-phase laying hens.

## 2. Materials and Methods

### 2.1. Sampling

Local breed chickens were locally sacrificed in different regions of China. The chicken intestines were separated and stored at −20 °C. The samples were then transported to our lab in dry ice, and the contents in duodenum, ileum, and cecum were collected in a super-clean table.

### 2.2. Isolation and Identification of Lactobacillus from Chicken Gut Contents

The procedure of isolation and identification was conducted according to Lee et al. [26] but with slight modifications. In addition, 0.1 g of chicken gut contents were suspended in 0.9 mL of sterile saline (0.85% *w*/*v*, pH 7.2–7.4) and homogenized with an electrical blender for 20 min. Serial decimal dilutions were prepared in the same diluent, and 0.1 mL was inoculated in triplicates by surface spreading on de Man, Rogosa, and Sharpe agar (MRS) containing 2% (*w*/*v*) CaCO_3_. Colonies that produced acid and formed clear halos were selected for further studies and streaked to purity on MRS Agar. For the identification of bacteria, colony PCR was performed with primers 27F (5′-AGAGTTTGATCCTGGCTCAG-3′) and 1492R (5′-ACGGCTACCTTGTTACGACTT-3′). The bacterial species was determined by BLAST searching the 16S rRNA gene sequences in the EZBioCloud database (https://www.ezbiocloud.net/) (accessed on 10 July 2021).

### 2.3. DNA Extraction

Each single colony was picked and cultured in normal MRS broth. For microbial genomic DNA extraction, 2 mL liquid cultures were centrifuged for 2 min at 8000× *g*. Total DNA from the pellets was extracted using a DNA extraction kit (CWBIO, Beijing, China) according to the manufacturer’s instructions.

### 2.4. In Vitro Study

#### 2.4.1. Acid and Bile Salt Tolerance Assay

This assay was tested according to Leite et al. [27], with slight modifications. In addition, 0.1 mL of an overnight culture of *Lactobacillus* strains (10^8^ CFU/mL) were inoculated into 5 mL of MRS broth previously adjusted to pH 3, or to MRS broth containing 0.3% or 0.5% (*w*/*v*) Oxgall (Solarbio, Beijing, China) to imitate gastric juice at 37 °C for 2 h. After incubation, the viability of cells was determined after 0, 2 h incubation by serial dilution and plating onto MRS agar. Isolates showing resistance more than 80% at pH 3 were considered acid tolerant strains. Survival rate (%) = Final (Log CFU/mL)/Initial (Log CFU/mL) × 100.

#### 2.4.2. Auto and Co-Aggregation Assay

Auto-aggregation capacity was done according to Todorov et al. [28]. Tested *Lactobacillus* strains, grown aerobically in MRS broth for 24 h in 37 °C, were harvested by centrifugation (8000× *g*, 5 min), then washed twice, resuspended, and diluted in PBS. The absorbance OD_600_ of bacterial suspension was adjusted in the range of 0.25 ± 0.05. The initial absorbance value was recorded as A_0_. After a 2-h incubation at 37 °C, the OD_600_ value was measured again as A_t_. Auto-aggregation was determined using the equation: auto-aggregation = [(A_0_ − A_t_)/A_0_] × 100%.

For the evaluation of co-aggregation capacity, the *Lactobacillus* strains were similarly grown in MRS broth, harvested by centrifugation (8000× *g*, 5 min), washed, resuspended, and diluted. Each *Lactobacillus* suspension was mixed with the same volume of a suspension of the coaggregation partner such as *Escherichia coli* CAU0757 (*E.*
*coli*), *Salmonella typhimurium* CMCC50115 (*S. typhimurium*), and *Clostridium perfringens* ATCC 13124 (*C. perfringens*), respectively, which were vortexed for 30 s. The determination method is the same as before.

#### 2.4.3. Hydrophobicity Assay

The cell surface hydrophobicity of *Lactobacillus* strains, as a measure to evaluate adherence to hydrocarbons, was performed according to the method of Abbasiliasi et al. [29]. Three tubes, each containing 3 mL of each lactic acid bacteria (LAB) strain suspension in PBS (pH 7.2) at 10^8^ CFU/mL, were mixed with the solvent n-hexadecane (1 mL) and vortexed for 1 min. The mixture was subsequently allowed to separate into 2 phases by standing for 5 to 10 min, after which the OD (at 600 nm) of the aqueous phase was measured with a spectrophotometer. Bacterial affinity to the solvent (hydrophobicity) was expressed using the formula (1 − A_10min_/A_0min_) × 100%, where A_10min_ and A_0min_ are the absorbance at 10 and 0 min, respectively.

#### 2.4.4. Antibacterial Ability Test

Antimicrobial activity of the isolates against pathogenic bacteria viz. *E. coli*, *S. typhimurium*, and *C. perfringens* was determined by the Oxford Cup method in accordance with Muhammad et al. [30]. Briefly, 100 μL solution of pathogenic bacteria cultured overnight were plated onto LB agar. After the plate was dried, 4 Oxford cups were put onto the plate equidistantly with 200 μL test strains culture solution in each cup at 37 °C for 24 h. The diameter of the clear zones was scored as follows: less and equal 10 mm (weak, +), 10–15 mm (strong, ++), and more than 15 mm (very strong, +++). The Ampicillin and non-cultured MRS broth were used as positive and negative controls.

#### 2.4.5. Hemolytic Activity

To be safe for use, probiotic strains must be nonhemolytic. All *Lactobacillus* strains tested were plated on Columbia agar, containing 5% (*w*/*v*) sheep blood, and incubated for 48 h at 37 °C. Blood agar plates were examined for signs of β-hemolysis (clear zones around colonies), α-hemolysis (green-hued zones around colonies), or γ-hemolysis (no zones around colonies) [31].

### 2.5. Genomic and Phylogenetic Analysis of the Selected Strain

The morphology of the strain was photographed and observed under scanning electron microscope. To evaluate the diversity of the selected *Lactobacillus* strain and the phylogenetic links to other *Lactobacillus*, sequences of the selected strain and 18 closed *Lactobacillus* strains were downloaded from the NCBI GenBank database (http://www.ncbi.nlm.nih.gov/genome/) (accessed on 19 December 2021); then, a phylogenetic tree was constructed using the iTOL (https://itol.embl.de/tree/120218717495301641453059) (accessed on 19 December 2021). Cluster of Orthologous Groups of proteins database (COG) was compared using BLASTP to identify the functional groups. BLAST Ring Image Generator (BRIG) was used to draw a genome map based on the JAVA platform. For comprehensive genetic analysis, an assembled genome was additionally annotated by RAST (https://rast.nmpdr.org/) (accessed on 21 December 2021). Database of Antimicrobial Activity and Structure of Peptides v3.0 (DBAASP, https://dbaasp.org/home (accessed on 1 March 2022) was used to predict the produced antimicrobial peptides (AMPs) by strain CML352.

### 2.6. Fermentation and Vacuum Lyophilization of Lactobacillus Powder

*Lactobacillus* fermentation and powder preparation were carried out in accordance with the method used by Chen et al. [32]. *Lactobacillus* strain was cultured in MRS broth at 37 °C for 24 h. After incubation, culture was centrifuged at 3000× *g* for 30 min at 4 °C, after which the pellets were washed with distilled water, in which 10% degreased milk powder was added as a protective agent. The mixture was then frozen dry and grinded into powder. In the end, the concentration of bacterial powder determined by serial dilution and plating onto MRS agar. The viability of the freeze-dried bacteria was nearly 1 × 10^12^ CFU/g and was stored at −80 °C until further use.

### 2.7. In Vivo Study

#### 2.7.1. Animal Husbandry and Feeding Conditions

The experiment was carried out in accordance with the Chinese guidelines for animal welfare. In addition, 160 Hyline Brown laying hens at the age of 65 week were randomly divided into 2 dietary treatments according to dietary supplementation with 0 (Control), 1 × 10^8^ CFU of the *Lactobacillus salivarius* CML 352 per kg of feed. The formula of basal diet and its nutrient contents are shown in Table 1. Each treatment consisted of 8 replicates with 10 laying hens (5 cages per replicate, 2 laying hens per cage). Before the start of the experiment, all hens were fed with experimental diets for 4 weeks. All birds were provided with ad libitum water and feed, as well as similar environmental and hygienic conditions throughout the experimental period. All experimental procedures involving animals were approved by the Institutional Animal Care and Use Committee of China Agricultural University.

#### 2.7.2. Sample Collection

At the end of the experiment, 16 birds in total randomly selected of each replicate group were fasted for 12 h and weighed. Five eggs per replicate were collected for egg quality determination. The ileum segments were carefully dissected and rinsed with sterilized saline and the cecal chyme was taken. All the samples were placed in liquid nitrogen immediately and stored at −80 °C till further analysis. Blood samples were collected (4 to 5 mL) after the hens were sacrificed. The fresh blood was kept standing for 60–120 min at 37 °C. Serum was obtained by centrifugation at 3000 rpm for 15 min and then stored at − 80 °C until use.

#### 2.7.3. Laying Performance and Egg Quality

Egg production and egg weight were recorded daily by replicate and feed consumption for each replicate was weighed every week. Feed conversion ratio (FCR) was calculated as grams of feed consumption/egg weight for each replicate. As for egg quality measurement, the thickness of eggshells was measured by vernier caliper. Breaking strength, Haugh unit values, albumen height, and yolk color were determined by an Egg Analyzer (Israel Orka Food Technology Ltd., Ramat Hasharon, Israel).

#### 2.7.4. The Analysis of Cecal Microbiota

Alpha diversity indices (including Chao1, Ace, Shannon, and Simpson index) were calculated to evaluate microbial species evenness on MicrobiomeAnalyst (https://www.microbiomeanalyst.ca/) (accessed on 25 December 2021). Beta diversity was evaluated by principal coordinate analysis (PCoA) based on the unweighted UniFrac distance [33]. Taxa abundances at the phylum and genus levels were statistically compared between the two groups. Linear discriminant analysis (LDA) combined effect size measurements (LEfSe) were used to identify the differences in microbial composition online in the Galaxy (http://huttenhower.sph.harvard.edu/galaxy/) (accessed on 25 December 2021) [34]. Phylogenetic Investigation of Communities by Reconstruction of Unobserved States 2 (PICRUSt2) software was further used to predict the functional pathway from the 16S rRNA data [35]. STAMP version 2.1.1.0 was also used as a graphical tool.

#### 2.7.5. Assay of Antioxidant Enzymes in Serum

The serum was used for the measurement of antioxidant indexes. Catalase (CAT), total superoxide dismutase (T-SOD), glutathione peroxidase (GSH-Px), malondialdehyde (MDA), and the total antioxidant capacity (T-AOC) were measured using commercial kits (Jiancheng Bioengineering Institute, Nanjing, China) according to the manufacturer’s instructions.

#### 2.7.6. Quantitative RT-PCR Analysis of Gene Expression

Total RNA was extracted from ileum (50 mg) using Trizol reagent (Beyotime, Beijing, China) according to the manufacturer’s instructions. After extraction, complementary DNA (cDNA) was synthesized from 1 μg of total RNA using MMLV reverse transcriptase (TaKaRa, Dalian, China). Transcriptional changes were then identified by quantitative PCR, which was performed using the Premix Ex TaqTM with SYBR Green (TaKaRa, Dalian, China). Synthesized cDNA was stored at −20 °C until use. Expression levels of the following genes were analyzed by real-time quantitative PCR, which were listed in Table 2. The PCR conditions were as follows: pre-denaturation at 95 °C for 2 min for 1 cycle; followed by denaturation at 95 °C for 15 s and annealing at 60 °C for 15–30 s, repeat the above steps for a total of 40 cycles; with a melting curve at 95 °C for 15 s, 60 °C for 15 s and 95 °C for 15 s. Expressions of the genes were quantified with a BioRad CFX96 Sequence Detection System and TransStart Top Green qPCR SuperMix (TransGen Biotech, Beijing, China). mRNA was analyzed for relative expression of the genes mRNA contents after normalizing for β-actin.

### 2.8. Statistical Analysis

Statistical analysis was performed using SPSS Statistics 18.0 software. One-way (in vitro experiment) ANOVA and Student’s *t*-test (in vivo experiment) were used to calculate significance. The data were expressed as mean ± SEM. Differences were considered statistically significant if *p* ≤ 0.05.

## 3. Results

### 3.1. Isolation and Identification of Lactic Acid Bacteria from Chinese Local Breed Chickens

A total of 57 LAB strains, belonging to 18 different bacterial species, were isolated from fecal samples of Hy-line brown and 16 locally reared Chinese local chicken breeds (Table 3). In addition, 18, 6, and 6 strains were isolated from Hy-line brown, Yuqingxiang and Tuer chickens, respectively, which were the top three chicken breeds with the greatest number of LAB isolates. All the isolates were identified by aligning their 16S rRNA gene sequences with those in Ezbiocloud database, and these strains showed 99.71% to 100% 16S rRNA gene nucleotide identity with known LAB. *L. salivarius* were the most prevalent LAB in these chickens (15/57 isolates; 26%), followed by *Ligilactobacillus agilis* (9/57 isolates; 16%), probably suggesting that these two species were relatively dominant LAB in chicken gut. The following 11 *Lactobacillus* species were rare and were found only once in one chicken breed: *Limosilactobacillus mucosae*, *Limosilactobacillus panis*, *Lactobacillus kitasatonis*, *Limosilactobacillus pontis*, *Lactobacillus aviarius subsp. Aviarius*, *Limosilactobacillus vaginalis*, *Limosilactobacillus coleohominis*, *Lactobacillus helveticus*, *Limosilactobacillus ingluviei*, *Limosilactobacillus oris*, and *Limosilactobacillus frumenti*.

We first tested the tolerance of the isolates to low pH and low bile salt concentrations, which are important indicators for the strains’ ability to survive in the host’s gastrointestinal tract. The results showed that less than 20 isolated *Lactobacillus* spp. displayed a satisfactory surviving rate of more than 80% after exposure to pH 3 for 2 h (Table 4). These were strains of the species *L. saerimneri* (CML344, CML398), *L. salivarius* (CML350, CML352, CML346, CML345, CML348), *L. johnsonii* (CML319, CML312, CML333), and *L. agilis* (CML313, CML104, CML341 CML323). Among them, eight strains were found to possess a remarkable surviving capacity of above 90%; CML319, CML313, CML350, CML352, and CML346 were the top five strains showing the highest pH tolerance. These 20 isolates were further evaluated their ability to survive in low bile salt concentrations (Table 5). We found that only 8 out of 20 isolates showed resistance to both 0.3% and 0.5% bile salts, in which CML319 (*L. johnsonii*) showed strongest tolerance compared to others under 0.3% bile salt condition, while CML352 (*L. salivarius*) performed better under 0.5% bile salt condition.

We then examined the auto-aggregation and co-aggregation abilities of six LAB strains selected based on their ability to tolerate pH and bile salts. The auto-aggregation ability of LAB stains makes an important contribution to the adhesion of intestinal cells and to avoid the colonization of pathogens, while the co-aggregation ability of bacteria eliminates gastrointestinal pathogens by preventing them from adhering to host tissues [36]. The auto-aggregation abilities of these isolates ranged from 78.55 to 84.70% with CML352 showing the highest (84.70%) (Table 6). The co-aggregation of the LAB strains against *E. coli*, *S. typhimurium*, and *C. perfringens* were between 47.12% and 48.49%, 47.92% and 61.72%, and 45.51% and 56.38%, respectively. CML319, CML350, and CML104 displayed the highest abilities against the three pathogens, respectively. A further investigation on the cell surface hydrophobicity activities of these strains indicated that CML352 had the highest resistance (57.66%), followed by CML350, CML344, CML398, CML319, and CML104 with 50.37, 49.08, 41.66, 41.13, and 21.21%, respectively (Table 6).

Lastly, we evaluated the direct inhibitory effects of the six isolates on three microbial pathogens, and, for safety reasons, the strains’ hemolytic activities. The results showed that all the six strains inhibited the growth of *C. perfringens*, whereas their inhibitory abilities against *E. coli* and *S. typhimurium* were relatively lower (Table 7). The highest zone of inhibition (≥ 20 mm) against *C. perfringens* was observed with the isolates CML350, CML352, CML398, and CML344. CML319, CML350, CML352, and CML104 displayed slight α-type hemolysis, while CML344 and CML398 showed γ-type hemolysis. Taking all the probiotic characteristics of these isolates into account, we selected CML352 for further study because of its balanced probiotic features.

### 3.2. Genomic and Phylogenetic Analysis of L. salivarius CML352

*L. salivarius* CML352 was isolated from Shanzhongxian chicken, a local breed in Anhui, China (Table 3). It is a short rod-shaped bacterium with a size of approximately 1.5 μm long by 0.6 μm wide (Figure 1A). We sequenced the genome of *L. salivarius* CML352, and a draft genome consisted of 172 contigs with an N50 value of 69,089 bp were obtained. The genome included 2,094,457 bp with a 32.6 % G + C content; 2121 coding sequences (CDSs) and 63 RNA genes were predicted (Figure 1B). A phylogenetic tree including strain CML352 and other 18 *L. salivarius* isolates in database was constructed based on the whole genome SNPs (Figure 1C). *L. salivarius* CML352 was found most closely related to *L. salivarius* JSWX5_1, a strain isolated from human fecal samples in China [37]. *L. salivarius* CML352 and JSWX5_1, together with two other strains, FHNXY73M9 and FGSYC2M4_2, clustered in the same clade in the phylogenetic tree.

We then compared the COG functional categories of these four isolates (Figure 1D). The major COG categories detected in the four strains belonged to “Information storage and processing” (mainly ([J] Translation, ribosomal structure, and biogenesis, [K] Translation, [L] Replication, recombination and repair) and “Poorly characterized” (mainly ([S] Function unknown). The COG categories of [I] Lipid transport and metabolism, and [Q] Secondary metabolites biosynthesis, transport, and catabolism were less abundant in CML 352 compared with the others, while categories of “Cellular processes and signaling” ([K] Transcription, [J] post-translational modification, protein turnover, chaperones, and [L] Replication, recombination and repair), “Cellular processes and signaling” ([D] Cell cycle control, cell division, chromosome partitioning, [V] Defense mechanisms) and “Metabolism” ([F] Nucleotide transport and metabolism) were enriched in CML 352.

We further annotated the genomes of CML 352 and its close relative JSWX5_1 using a Rapid Annotation using Subsystem Technology (RAST) server. The results showed that nearly the same number of genes were assigned to SEED subsystems in these two strains (Figure 1E and Appendix A). Compared with JSWX5_1, CML352 had more genes in Nucleosides and Nucleotides (84 vs. 73), Regulation and Cell signaling (13 vs. 8), and DNA Metabolism (68 vs. 50), but less genes in Virulence, Disease, and Defense (33 vs. 37), Protein Metabolism (111 vs. 114), Carbohydrates (121 vs. 146), and Amino Acids and Derivatives (97 vs. 108). Both two strains had no relevant genes in Photosynthesis, Motility and Chemotaxis, Secondary Metabolism, and Nitrogen Metabolism.

We next explored the specific genetic features of CML352 strain based on the functional roles in each subsystem (Appendix A). CML352 had three genes that were relevant to phage packaging machinery and phage tail fiber, while no phage-related genes were found in JSWX5_1. In the subsystem of Nucleosides and Nucleotides, 49 genes related to purine metabolism existed in CML352, but only 38 such genes were present in JSWX5_1, indicating that CML352 may have more potent ability in metabolizing purine. Moreover, CML352 had more genes in DNA Metabolism than JSWX5_1. The most different category was CRISPRs (Clustered Regularly Interspaced Short Palindromic Repeats), in which CML352 had 13 related genes and JSWX5_1 only possessed two, which, to some extent, means that CML352 has a stronger immune function than JSWX5_1. Interestingly, three genes correlated to Sex pheromones in *Enterococcus faecalis* and other Firmicutes existed in CML352, but none was found in JSWX5_1. These genes seemed to be related to the synthesis of certain enzymes and proteins that could alter signaling, social behavior, and evolution of plasmids. Additionally, we found 41 predicted AMPs encoded by CML352, among which, nine, nine and two were active against *E. coli*, *S. typhimurium* and *C. perfringens*, respectively (Appendix A), suggesting the antibacterial ability of CML352 were probably due to the specific AMPs.

We then added *L. salivarius* CML352 to the diet of late-phase laying hens and evaluated the effects on this strain on the bird’s gut microbiota, intestinal epithelial barrier, immunity, antioxidant activity, production performance, and egg quality.

### 3.3. L. salivarius CML352 Modulated the Gut Microbiota Composition and Function of Laying Hens

No significant differences of the alpha diversity of cecal microbiota, estimated by Shannon, Simpson, Ace, and Chao1 indices, were found between the two groups (*p* > 0.05). However, beta diversity results (PCoA analysis) indicated that the microbiota of *L. salivarius* CML352 group was clearly different from the control group (*p* < 0.05) (Figure 2A,B). Overall, the microbiota was dominated by the phylum Firmicutes and Bacteroidetes in the two groups. Dietary *L. salivarius* CML352 supplementation significantly decreased the relative abundance of phylum Firmicutes, while increasing the relative abundance of phylum Bacteroidetes (*p* < 0.05), thus leading to a significant decrease in Firmicutes to Bacteroidetes ratio (*p* < 0.05) (Figure 2D). The differentially abundant operational taxonomic units (OTUs) at the phylum and genus levels were further analyzed by LEfSe (LDA > 2.0; Figure 2E,F). As shown in Figure 2E, Bacteroidetes and Epsilonbacteraeota were enriched in *L. salivarius* CML352-treated group, while Firmicutes was the enriched phylum in the control group. At the genus level, *Romboutsia*, *Ruminococcaceae UCG-013*, *Blautia*, *Ruminiclostridium 9*, *Ruminococcaceae UCG-004*, *Marvinbryantia*, *Ruminiclostridium 5*, *Lachnospiraceae UCG-010*, *CAG-56*, and *GCA-900066575* were enriched in *L. salivarius* CML352 group (Figure 2F).

The PICRUSt2 software was further used to predict the microbial functional categories and eight differentially abundant MetaCyc pathways were found between the two groups (q < 0.05, Figure 3). Among them, the microbial gene functions related to GDP-mannose biosynthesis were much higher in the fecal microbiome of the CML352 group, which has been recently proposed to play a vital role in the biosynthesis of Vitamin C as a valuable antioxidant [38]. In contrast, the abundance of pathways related to lipid synthesis were decreased in the CML352 group, such as phosphatidylglycerol biosynthesis II (non-plastidic), the super pathway of phospholipid biosynthesis I (bacteria), CDP-diacylglycerol biosynthesis I, and CDP-diacylglycerol biosynthesis II. In addition, the gene functions associated with methanogenesis from acetate in CML352 group were significantly lower than the control group, which has been presumed to reduce fat deposition and obesity [39].

### 3.4. Effects of L. salivarius CML352 on Intestinal Epithelial Barrier, Immunity, and Antioxidant Activity

No statistically significant differences (*p* > 0.05) were observed on the relative mRNA expression of zonula occludens-1 (*ZO-1*), *Occludin*, and *Claudin-2* in ileal mucosa of laying hens in response to the addition of *L. salivarius* CML352 (Figure 4A). However, there was a significant increase (*p* < 0.05) in relative mRNA expression of mucin-2 (*Muc-2*) in the ileum. In addition, *L. salivarius* CML352 tended to decrease the relative mRNA expression of *Claudin-2* (*p* < 0.1).

Dietary supplementation with *L. salivarius* CML352 downregulated (*p* < 0.05) the relative mRNA expression of myeloid differentiation factor88 (*MyD88*), interferon-γ (*IFN-γ*) and toll-like receptor-4 (*TLR-4*) in the ileum, while having no obvious impact on tumor necrosis factor-α (*TNF-α*), transforming growth factor-β (*TGF-α*), nuclear factor kappa-B (NF-κB), interleukin-2 (*IL-2*), and interleukin-1β (*IL-1β*) levels (Figure 4B).

Compared to the control group, *L. salivarius* CML352 supplementation tended to reduce the concentrations of MDA in serum (*p* < 0.1), whereas a slight but not significant increase in GSH-Px and T-SOD concentrations was observed in the treated group (*p* < 0.1). There was no difference in T-AOC and CAT activity (*p* > 0.05). Levels of serum T-AOC were similar to that in the control group. In addition, the values of CAT activity were not significantly affected by *L. salivarius* CML352 treatment (Figure 4C).

### 3.5. Effects of L. salivarius CML352 on Production Performance and Egg Quality

In comparison with the control, dietary *L. salivarius* CML352 supplementation had no obvious effect (*p* > 0.05) on egg production and feed conversion ratio of laying hens during the whole experiment (Table 8). However, the average egg weight was significantly increased (*p* < 0.05) during weeks 56–67, and a positive trend was observed during weeks 56–59 and 60–63 (*p* < 0.1). Meanwhile, *L. salivarius* CML352 addition significantly decreased average daily feed intake during weeks 64–67 (*p* < 0.05). For the entire experimental period, dietary supplementation with *L. salivarius* CML352 resulted in a slight reduction in the average weight of laying hens (*p* < 0.1), while the abdominal fat deposition was significantly reduced in the treated group (*p* < 0.05) (Table 9).

With regard to egg quality, there was no apparent difference (*p* > 0.05) on yolk percentage, and albumen height at the end of the experiment, while a significant increase on eggshell strength and eggshell thickness (*p* < 0.05) and an increasing but not significant trend for Haugh units were observed (Table 10).

## 4. Discussion

### 4.1. Probiotic Properties Assessment

In this study, *Lactobacillus* strains were isolated from chicken fecal samples of Chinese local breed chickens in different regions, and the probiotic properties of selected strains were investigated. Even though the bacteria belonging to *Lactobacillus* species are common probiotics and are ‘generally recognized as safe’ (GRAS), the probiotic properties of LAB have been demonstrated to be strain-specific, and in vitro assessment of newly isolated strains is necessary [40]. Thus, 57 *Lactobacillus* isolates were tested for the simulated gastrointestinal tolerance, the antimicrobial effect, the co-aggregation ability and hydrophobicity, and the hemolysis activity.

Acid and bile salt tolerance are frequently considered during the selection of potential probiotic strains to guarantee their viability and functionality [41]. It is proposed that the degree of acid or bile salt tolerance of probiotics is highly variable and strain-dependent [42]. Plenty of studies have discussed the acid and bile salt environment tolerance of probiotics [43,44], and it has been confirmed that LABs could survive better in simulated intestinal environment than non-LAB strains [44]. Previous study showed that five out of twelve LAB isolates were detected to resist to acid pH value 3 and 0.45% (*w*/*v*) of bile salt, and the maximum resistance was 60.52% and 68.62%, respectively [45]. Similar results were observed in this study that only eight strains (one *L. johnsonii*, two *L. agilis*, three *L. salivarius*, and two *L. saerimneri* isolates) showed high tolerance to both the acid and bile salt environments, suggesting their potential ability to survive the harsh conditions in the real stomach and small intestinal tract. Adherence and colonization of the intestine are also important probiotic functions, which could suppress and exclude pathogenic microbes, and are essential for immunomodulatory activity [46]. The adhesive ability is closely related to the strain’s hydrophobicity and auto-aggregation capacity [47,48]. Additionally, the co-aggregation capability is demonstrated to prevent the GIT of the host from being colonized by pathogens [36]. All the tested strains in our study showed different degrees of adhesive properties, among which CML352 exhibited the highest auto-aggregation (84.75%) and hydrophobicity (57.66%) capabilities. Interestingly, CML352 also showed a strong inhibition against *E. coli*, *S. typhimurium*, and *C. perfringens* in vitro. As opposed to α-hemolysis, β-hemolytic activity is considered harmful. Strains exhibiting this capability are known to produce exotoxin, which lyses blood cells and therefore impacts the immune system [49]. Non-enterococcal LABs exhibiting α-hemolytic may harbor low virulence potential and have been regarded to be safe to use [26,50,51]. None of the strains in this study showed β-hemolysis, indicating that the selected strains were relatively safe to use. Collectively, we proposed that *L. salivarius* CML352 is a candidate probiotic that can survive the intestinal conditions, inhibit gut pathogens, and thus exert beneficial effects on laying hens. We should stress that, in the present study, we didn’t cover all probiotic features and safety concerns of a potential probiotics, such as resistance to lysozyme [52,53] and antibiotic susceptibility [54], which should be further considered and evaluated in the future work.

### 4.2. Effect of L. salivarius CML352 on Cecal Microbiota of Layers

Dietary supplementation of *L. salivarius* CML352 did not significantly influence the alpha diversity of the cecal microbiota of late-phase laying hens, but had a remarkable impact on the beta diversity. Studies have shown that *Lactobacillus* addition could increase cecum microbial diversity [55]. A significant reduction of Firmicutes and increase of Bacteroidetes were found after the CML352 treatment, thus inducing a sharply reduced ratio of Firmicutes to Bacteroidetes (F/B) in the chicken intestinal microbiota. Numerous human studies have consistently demonstrated that the F/B proportion is increased in obese people compared to lean people, and tend to decrease with weight loss [56]. In chicken, it is also demonstrated that the F/B ratio was positively correlated with the accumulation of chicken abdominal fat deposition [57]. This might be one of the reasons to explain the significant decrease of abdominal proportion and the positive trend of decreased average body weight in the CML352 group. The reduced abundance of Firmicutes phylum in the chicken gut may diminish the bacterial species with the ability to harvest energy from diet, thus leading to a large decrease in total body fat [58].

We further investigate the role of *L. salivarius* CML352 in chicken gut by analyzing the changes of microbial metabolic pathways. It was revealed that the most significantly reduced microbial pathways in the CML352 group were involved in the lipid synthesis. In addition, the gene functions associated with methanogenesis from acetate in CML352 group were significantly lower than the control group. Studies have demonstrated that methane could slow down the peristalsis of intestinal and increase the digestion time, thus improving the absorption of nutrients and inducing obesity [39]. Taken together, we suggested that dietary supplementation of *L. salivarius* CML352 in laying hens can modulate the gut microbiota by decreasing the F/B ratio, reducing lipid synthesis-related pathways and lowing methanogenesis from acetate, and thus reducing the abdominal fat deposition in layers.

In addition, the increase of Bacteroidetes may benefit the chicken gut health, as most species in Bacteroidetes produce short-chain fatty acids (SCFAs) as their final fermentation products [59]. The other CML352-enriched bacteria such as *Phascolarctobacterium* and *Odoribacter* were also SCFA-producers [60,61]. SCFAs play an important role in innate immunity as they have anti-inflammatory effects [62], which affects both nonimmune as well as immune intestinal cells and modulates intestinal homeostasis [63]. Additionally, SCFAs also serve as a vital energy source for the intestinal epithelial cells [64], and have important intestinal and immuno-modulatory function [65], and an influence on maintaining epithelial barrier function and suppressing proinflammatory cytokines [66].

### 4.3. Effect of L. salivarius CML352 on Intestinal Epithelial Barrier, Immunity, and Antioxidant Activity

The intestinal epithelium is an integral component of the intestinal barrier and provides the first line of defense against external stimuli [67], which could maintain intestinal homeostasis [68]. The integrity of the gut epithelium is maintained by the formation of tight junction protein complexes between cells. Studies in various animals have shown the beneficial effects of probiotics on the regulation of tight junction proteins, which could increase host epithelial barrier and protect against pathogen invasion into internal organs [69]. In this study, we observed a decreasing trend of mRNA expression of *Claudin-2* (*p* < 0.1). Tight junction protein *Claudin-2* is considered to be associated with the expression of cation-selective pores, which may increase paracellular permeability [70]. For example, an increase of *Claudin-2* induced by *S. typhimurium* infection increased mucosal permeability, thus disrupting the intestinal barrier in chicks [71]. Mucins are protective, antimicrobial substances secreted by epithelial cells, among which *Muc-2* is the primary gel-forming mucin in the mammalian gut [72], playing an essential role in preventing pathogenic bacteria from destroying enterocytes [73]. Our data indicated that dietary supplementation of *L. salivarius* CML352 significantly upregulated the expression of *Muc-2*. It is reported that microbial-derived SCFAs could boost the expression of tight junction proteins and repress paracellular permeability [74] through stimulating the goblet cells to secrete mucin, especially *Muc-2*. Thus, we suggest that the high expression of *Muc-2* in *L. salivarius* CML352 group was probably due to the resulting enrichment of the SCFAs-producers. Taken together, dietary supplementation of *L. salivarius* CML352 decreased intestinal permeability and improved intestinal health of layers.

Studies have demonstrated that *Lactobacillus* or their cell components can improve immune function in animals [55]. For example, *L. acidophilus* supplementation enhanced the cellular and humoral immunity in *E. coli* challenged chickens [75]. In addition, probiotic *Lactobacillus* strains modulated the gut environment and stimulated the immune system in layer- and meat-type chickens [76]. This research demonstrated that *Lactobacillus* can increase host immunity with or without the challenge of pathogens. Here, we showed that supplementation of *L. salivarius* CML352 significantly reduced *MyD88* and *TLR4* expression, which were speculated to be associated with the underlying anti-inflammatory effect [77]. We also found that the level of mucosal pro-inflammatory cytokines *IFN-γ* was significantly reduced in the *L. salivarius* CML352-treated group, consistent with previous evaluation of the effects of *L. salivarius* on chickens [78]. These results indicated that the administration of *L. salivarius* CML352 decreased the inflammatory level and maintained immune homeostasis in layers.

It is reported that *L. salivarius* showed good anti-oxidative properties. For example, *L. salivarius* supplementation effectively alleviated the organ damage and enhanced anti-oxidative capacity in both acute heat stress and circular heat stress condition in chicken [79]. Malondialdehyde is a biomarker of lipid peroxidation and its level is directly proportionate to the extent of the cellular damage caused by free radicals [80]. We found that, when *L. salivarius* CML352 was added to the diet, a trend of reduction of MDA activity in the serum of layers occurred (*p* = 0. 079). Additionally, CML352 resulted in an increasing trend of GSH-Px (*p* = 0.075) and T-SOD (*p* = 0.090) in serum. These results suggested that *L. salivarius* CML352 can improve serum antioxidant status of layers. Similarly, the use of other *Lactobacillus* strains, i.e., *L. plantarum* JM113, also resulted in the greater SOD and GSH-Px activities and the decreased MDA activity in chickens [81]. It is proposed that the Nrf2-ARE signaling pathway was involved in the protective effect of *L. plantarum* and regulates the expression and activity of antioxidative enzymes [82]. Whether or not Nrf2-ARE signaling pathway mediated the anti-oxidative capacity of *L. salivarius* CML352 needs to be further investigated.

### 4.4. Effect of L. salivarius CML352 on the Production Performance of Layers

Our results showed that the average egg weight was significantly increased (*p* < 0.05) during the entire experiment, and a positive trend was observed during weeks 56–59 and 60–63 (*p* < 0.1). Meanwhile, *L. salivarius* CML352 addition significantly decreased average daily feed intake during weeks 64–67 (*p* < 0.05). A limited number of studies have investigated the effects of dietary supplementation with *L. salivarius* on the performance of laying hens. However, increased egg weight or improved feed efficiency in hens have been observed by using other *Lactobacillus* strains [83], and dietary administration of *L. acidophilus* at 2 × 10^9^ and 3 × 10^9^ CFU/kg significantly increased the average egg weight by 1.7 and 1.9 g, respectively, and, at the same time, reduced the daily feed consumption [17]. Similarly, various combinations of *Lactobacillus* species (*L. rhamnosus*, *L. paracasei*, and *L. plantarum*) significantly improve the egg production [84]. These beneficial effects may be attributed to the improvement of nutrient digestibility in the daily diet by *Lactobacillus* and thus increased the efficiency of food utilization and resulted in heavier eggs. Furthermore, a significant increase on eggshell strength and eggshell thickness (*p* < 0.05) were observed after the administration of *L. salivarius* CML352. This may be because *Lactobacillus* can produce a variety of vitamins in the intestinal tract of poultry and promote the absorption of vitamins and mineral elements [85]. Additionally, we observed an increasing but not significant trend for Haugh units (*p* < 0.1), a measurement of the internal quality of an egg, which is also supported by other studies using *L. salivarius* as a feed additive [86]. Collectively, our findings, together with results in previous studies, confirmed the role of *L. salivarius* in improving laying hens’ production performance and egg quality.

## 5. Conclusions

In summary, we isolated and characterized the probiotic properties of *Lactobacillus* strains from Chinese local breed chickens. We sequenced the genome of the newly isolated *L. salivarius* strain, CML352, and demonstrated the beneficial effects of this probiotic candidate on modulating the layers’ gut microbiota, abdominal fat deposition, immunity, antioxidant capacity, and production performance. Results gained so far indicated that our strain may have a potential effect on the laying hen industry, but a further investigation of the mechanisms involved is still needed to warrant a future application of this bacterium.

## Figures and Tables

**Figure 1 microorganisms-10-00726-f001:**
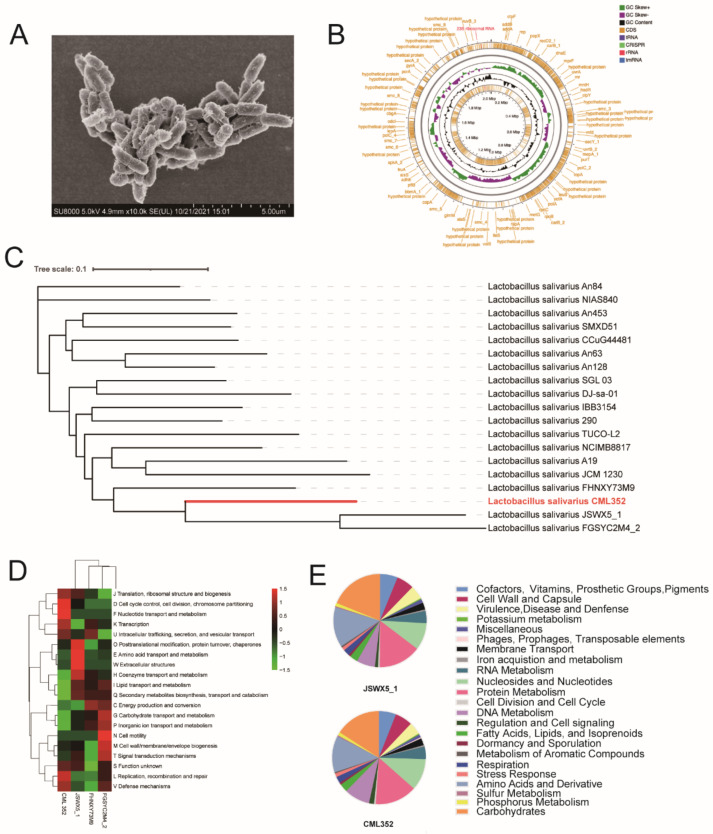
(**A**) The SEM picture of CML 352, (**B**) the Genome map of CML 352, (**C**) phylogenetic tree of CML 352, (**D**) the clusters of orthologous groups (COG) repertoires, (**E**) the subsystem category distribution of JSWX5_1 and CML 352.

**Figure 2 microorganisms-10-00726-f002:**
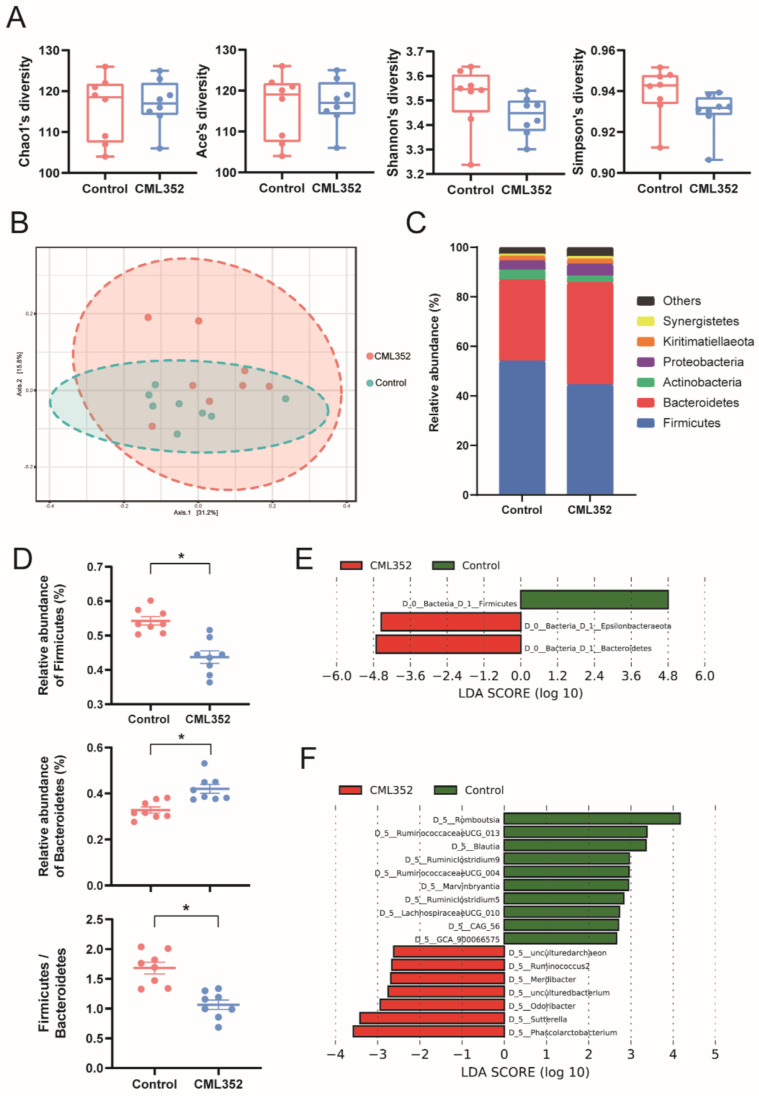
Influence of dietary *L. salivarius* CML352 supplementation on the gut microbiota composition. (**A**) four different alpha diversity metrics, (**B**) PcoA 2D Graphics generated from MicrobiomeAnalyst, where red circles represent abundance in hens fed with *L. salivarius* CML352, blue circles for those of hens in the control group, (**C**) relative abundance of bacterial phyla detected in the samples. Those abundance below 1% were classified into “others”, (**D**) relative abundance of Firmicutes and Bacteroidetes phyla, and the ratio of Firmicutes to Bacteroides, asterisks indicated values significantly different (*p* < 0.05) between the groups, (**E**) the LEfSe analysis of the cecum microbiota at the phylum level, and (**F**) the LEfSe analysis of the cecum microbiota at the genus level.

**Figure 3 microorganisms-10-00726-f003:**
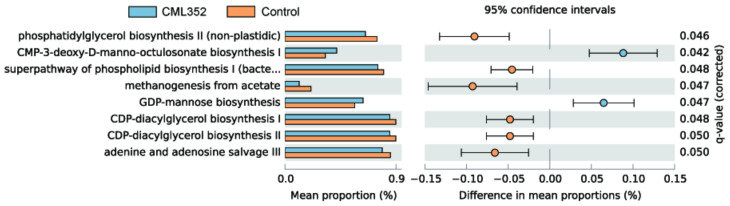
PICRUSt analysis in the MetaCyc pathways; functional predictions for the fecal microbiome of the *L. salivarius* CML352 group and the control group. Significant MetaCyc pathways for the fecal microbiome of the 2 groups were identified by STAMP software.

**Figure 4 microorganisms-10-00726-f004:**
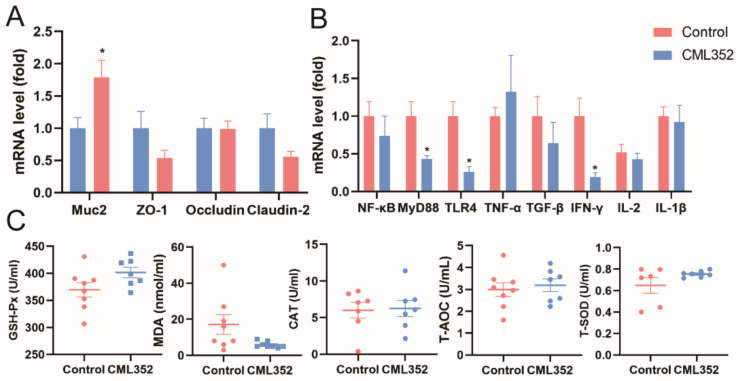
(**A**) Effects of *L. salivarius* CML352 supplementation on intestinal epithelial barrier in ileum of laying hens. Total RNA was extracted and the expressions of *Muc-2*, *ZO-1*, *Occludin* and *Claudin-2* and were measured by real-time PCR; (**B**) effects of *L. salivarius* CML352 supplementation on immunity in ileum of laying hens. Total RNA was extracted and the expressions of *NF-κB*, *MyD88*, *TLR-4*, *TNF-α*, *TGF-α*, *IFN-γ*, *IL-2*, and *IL-1β* were measured by real-time PCR; (**C**) effects of *L. salivarius* CML352 supplementation on serum antioxidant of laying hens. Data are expressed as mean ± SEM from three independent experiments. Asterisks indicated values significantly different (*p* < 0.05) between the groups.

**Table 1 microorganisms-10-00726-t001:** Composition and nutrient levels of the basal diet (air-dry basis, g/kg).

Ingredients	Calculated Nutrient Analysis		
	Ratio, %	Nutrient Levels	Content, %
Corn [7.8%] ^1^	63.81	Metabolizable energy (MJ/kg)	2.777
Soybean meal [43%] ^1^	20.0	Crude protein (%)	16.445
Limestone	8.3	Calcium (%)	3.5
Corn gluten meal [60%] ^1^	3.5	Available phosphorus (%)	0.365
Dicalcium phosphate	1.5	Lysine (%)	1.216
Soybean oil	1.3	Methionine (%)	0.441
L-lysine-HCl [78%] ^2^	0.649	Methionine + cystine	0.7
NaCl	0.35		
Mineral premix	0.25		
DL-methionine	0.185		
Choline chloride 50% ^2^	0.1		
Vitamin premix ^3^	0.02		
Ethoxyquin MAX ^4^	0.02		
Phytase 10,000 ^5^	0.016		
*Lactobacillus strain*	+/− ^6^		
Total	100		

^1^ Corn, soybean meal, and corn gluten meal with the protein proportion are 7.8%, 43%, and 60% respectively. ^2^ L-lysine-HCl and choline chloride with the effective substance proportion are 78% and 50%, respectively. ^3^ Vitamin premix (1 g): vitamin A 82.5 IU, vitamin E 160 mg, vitamin D3 12,000 U, vitamin K 10 mg. ^4^ Ethoxyquin MAX: feed antioxidant. ^5^ Phytase10,000: enzymes of feed grade with the specification are 10,000 U/g. ^6^
*Lactobacillus* was supplemented at 0 (control group), 2 g/kg (CML352 group).

**Table 2 microorganisms-10-00726-t002:** Gene name, primer sequences.

	Primer Sequence
*Muc2*	F: 5′ATTGTGGTAACACCAACATTCATC3′ R:5′CTTTATAATGTCAGCACCAACTTCTC3′
Occludin	F: 5′ACGGCAGCACCTACCTCAA3′ R: 5′GGGCGAAGAAGCAGATGAG3′
*ZO-1*	F: 5′GAATGATGGTTGGTATGGTGCG3′ R: 5′GGGCGAAGAAGCAGATGAG3′
Claudin-2	F: 5′TCAGAAGTGTGTCTACTGTCCG3′ R: 5′AACTCACTCTTGGGCTTCTG3′
*NF-κB*	F: 5′CTCTCCCAGCCCATCTATGA3′ R: 5′CCTCAGCCCAGAAACGAAC3′
*MyD88*	F: 5′AGAAGGTGTCGGAGGATGGT3′ R: 5′GCTGGATGCTATTGCCTCGCTG3′
*TNF-α*	F: 5′GGATGGATGGAGGTGAAAGTAG3′ R: 5′TGATCCTGAAGAGGAGAGAGAA3′
*TGF-β*	F: 5′TCATCACCAGGACAGCGTTA3′ R: 5′TGTGATGGAGCCATTCATGT3′
*IFN-γ*	F: 5′AAAGCCGCACATCAAACACA3′ R: 5′GCCATCAGGAAGGTTGTTTTTC3′
*IL-2*	F: 5′GCAGTGTTACCTGGGAGAAGTG3′ R: 5′TCTTGCATTCACTTCCGGTGT3′
*IL-1β*	F: 5′GAAGTGCTTCGTGCTGGAGT3′ R: 5′ACTGGCATCTGCCCAGTTC3′
*TLR4*	F: 5′CCACTATTCGGTTGGTGGAC3′ R: 5′ACAGCTTCTCAGCAGGCAAT3′
β-actin	F: 5′ACTGGCACCTAGCACAATGA3′ R: 5′CTGCTTGCTGATCCACATCT3′

F: forward primer, R: reverse primer. *Muc2*: mucin-2, *ZO-1*: zonula occludens-1, *NF-κB*: nuclear factor kappa-B, *MyD88*: myeloid differentiation factor88, TNF-α: tumor necrosis factor-α, *TGF-α*: transforming growth factor-β, *IFN-γ*: interferon-γ, *IL*: interleukin, TLR: toll-like receptors.

**Table 3 microorganisms-10-00726-t003:** Information of 57 LAB isolates from the chicken gut.

	Phylum	Genus	Species	Chicken Breed	Origin
*Lactobacillus salivarius*					
CML316	Firmicutes	*Lactobacillus*	*Lactobacillus* *salivarius*	Green Shell layer	Beijing
CML320	Firmicutes	*Lactobacillus*	*Lactobacillus* *salivarius*	817 chicken	Yunfu, Guangdong
CML322	Firmicutes	*Lactobacillus*	*Lactobacillus* *salivarius*	Aixiang chicken	Yunfu, Guangdong
CML327	Firmicutes	*Lactobacillus*	*Lactobacillus* *salivarius*	Tuer chicken	Yunfu, Guangdong
CML331	Firmicutes	*Lactobacillus*	*Lactobacillus* *salivarius*	Yuqingxiang chicken	Yunfu, Guangdong
CML338	Firmicutes	*Lactobacillus*	*Lactobacillus* *salivarius*	Mahuanggong chicken	Yunfu, Guangdong
CML342	Firmicutes	*Lactobacillus*	*Lactobacillus* *salivarius*	Jianghan chicken	Wuhan, Hubei
CML345	Firmicutes	*Lactobacillus*	*Lactobacillus* *salivarius*	Pingguo chicken	Jinan, Shandong
CML346	Firmicutes	*Lactobacillus*	*Lactobacillus* *salivarius*	Langya chicken	Langya, Shandong
CML348	Firmicutes	*Lactobacillus*	*Lactobacillus* *salivarius*	Luhua chicken	Wenshang, Shandong
CML350	Firmicutes	*Lactobacillus*	*Lactobacillus* *salivarius*	Shiqi chicken	Jinan, Shandong
CML351	Firmicutes	*Lactobacillus*	*Lactobacillus* *salivarius*	Shouguang chicken	Shouguan, Shandong
CML352	Firmicutes	*Lactobacillus*	*Lactobacillus* *salivarius*	Shanzhongxian chicken	Xuancheng, Anhui
CML354	Firmicutes	*Lactobacillus*	*Lactobacillus* *salivarius*	Bairi chicken	Jining Shandong
CML111	Firmicutes	*Lactobacillus*	*Lactobacillus* *salivarius*	Hy-line brown	Zhuozhou, Hebei
*Lactobacillus gallinarum*					
CML330	Firmicutes	*Lactobacillus*	*Lactobacillus gallinarum*	817 chicken	Yunfu, Guangdong
CML329	Firmicutes	*Lactobacillus*	*Lactobacillus gallinarum*	Tuer chicken	Yunfu, Guangdong
CML334	Firmicutes	*Lactobacillus*	*Lactobacillus gallinarum*	Yuqingxiang chicken	Yunfu, Guangdong
CML190	Firmicutes	*Lactobacillus*	*Lactobacillus gallinarum*	Hy-line brown	Zhuozhou, Hebei
*Ligilactobacillus agilis*					
CML313	Firmicutes	*Ligilactobacillus*	*Ligilactobacillus agilis*	Nongda third chicken	Beijing
CML318	Firmicutes	*Ligilactobacillus*	*Ligilactobacillus agilis*	817 chicken	Yunfu, Guangdong
CML326	Firmicutes	*Ligilactobacillus*	*Ligilactobacillus agilis*	Tuer chicken	Yunfu, Guangdong
CML323	Firmicutes	*Ligilactobacillus*	*Ligilactobacillus agilis*	Aixiang chicken	Yunfu, Guangdong
CML336	Firmicutes	*Ligilactobacillus*	*Ligilactobacillus agilis*	Yuqingxiang chicken	Yunfu, Guangdong
CML341	Firmicutes	*Ligilactobacillus*	*Ligilactobacillus agilis*	Jianghan chicken	Wuhan, Hubei
CML337	Firmicutes	*Ligilactobacillus*	*Ligilactobacillus agilis*	Mahuanggong chicken	Yunfu, Guangdong
CML104	Firmicutes	*Ligilactobacillus*	*Ligilactobacillus agilis*	Hy-line brown	Zhuozhou, Hebei
CML200	Firmicutes	*Ligilactobacillus*	*Ligilactobacillus agilis*	Hy-line Grey	Zhuozhou, Hebei
*Lactobacillus johnsonii*					
CML312	Firmicutes	*Lactobacillus*	*Lactobacillus johnsonii*	Nongda third chicken	Beijing
CML325	Firmicutes	*Lactobacillus*	*Lactobacillus johnsonii*	Tuer chicken	Yunfu, Guangdong
CML333	Firmicutes	*Lactobacillus*	*Lactobacillus johnsonii*	Yuqingxiang chicken	Yunfu, Guangdong
CML189	Firmicutes	*Lactobacillus*	*Lactobacillus johnsonii*	Hy-line brown	Zhuozhou, Hebei
*Limosilactobacillus reuteri*					
CML189	Firmicutes	*Limosilactobacillus*	*Limosilactobacillus reuteri*	Hy-line brown	Zhuozhou, Hebei
CML143	Firmicutes	*Limosilactobacillus*	*Limosilactobacillus reuteri*	Hy-line Grey	Zhuozhou, Hebei
CML321	Firmicutes	*Limosilactobacillus*	*Limosilactobacillus reuteri*	Aixiang chicken	Yunfu, Guangdong
CML328	Firmicutes	*Limosilactobacillus*	*Limosilactobacillus reuteri*	Tuer chicken	Yunfu, Guangdong
CML335	Firmicutes	*Limosilactobacillus*	*Limosilactobacillus reuteri*	Yuqingxiang chicken	Yunfu, Guangdong
*Lactobacillus crispatus*					
CML319	Firmicutes	*Lactobacillus*	*Lactobacillus crispatus*	817 chicken	Yunfu, Guangdong
CML324	Firmicutes	*Lactobacillus*	*Lactobacillus crispatus*	Tuer chicken	Yunfu, Guangdong
CML332	Firmicutes	*Lactobacillus*	*Lactobacillus crispatus*	Yuqingxiang chicken	Yunfu, Guangdong
CML343	Firmicutes	*Lactobacillus*	*Lactobacillus crispatus*	Jianghan chicken	Wuhan, Hubei
CML400	Firmicutes	*Lactobacillus*	*Lactobacillus crispatus*	Hy-line brown	Zhuozhou, Hebei
*Lactobacillus saerimneri*					
CML314	Firmicutes	*Lactobacillus*	*Lactobacillus saerimneri*	Nongda third chicken	Beijing
CML317	Firmicutes	*Lactobacillus*	*Lactobacillus saerimneri*	Green Shell layer	Beijing
CML344	Firmicutes	*Lactobacillus*	*Lactobacillus saerimneri*	817 chicken	Yunfu, Guangdong
CML398	Firmicutes	*Lactobacillus*	*Lactobacillus saerimneri*	Hy-line brown	Zhuozhou, Hebei
Others					
CML158	Firmicutes	*Lactobacillus*	*Limosilactobacillus mucosae*	Hy-line brown	Zhuozhou, Hebei
CML174	Firmicutes	*Lactobacillus*	*Limosilactobacillus panis*	Hy-line brown	Zhuozhou, Hebei
CML176	Firmicutes	*Lactobacillus*	*Lactobacillus* *kitasatonis*	Hy-line brown	Zhuozhou, Hebei
CML177	Firmicutes	*Lactobacillus*	*Limosilactobacillus pontis*	Hy-line brown	Zhuozhou, Hebei
CML180	Firmicutes	*Lactobacillus aviarius*	*Lactobacillus* *aviarius subsp. aviarius*	Hy-line brown	Zhuozhou, Hebei
CML184	Firmicutes	*Lactobacillus*	*Limosilactobacillus vaginalis*	Hy-line brown	Zhuozhou, Hebei
CML185	Firmicutes	*Lactobacillus*	*Limosilactobacillus coleohominis*	Hy-line brown	Zhuozhou, Hebei
CML187	Firmicutes	*Lactobacillus*	*Lactobacillus* *helveticus*	Hy-line brown	Zhuozhou, Hebei
CML188	Firmicutes	*Lactobacillus*	*Limosilactobacillus ingluviei*	Hy-line brown	Zhuozhou, Hebei
CML202	Firmicutes	*Limosilactobacillus*	*Limosilactobacillus* *oris*	Hy-line brown	Zhuozhou, Hebei
CML193	Firmicutes	*Lactobacillus*	*Limosilactobacillus frumenti*	Hy-line brown	Zhuozhou, Hebei

**Table 4 microorganisms-10-00726-t004:** Survival rate of different *Lactobacillus* strains under pH = 3.

Scheme 0.	Species	0 h Living Bacteria (lg CFU/mL)	2 h Living Bacteria (lg CFU/mL)	Survival Rate (%)
CML319	*Lactobacillus johnsonii*	5.15 ± 0.04	5.06 ± 0.03	98.12
CML313	*Ligilactobacillus agilis*	5.86 ± 0.06	5.62 ± 0.09	95.87
CML350	*Lactobacillus salivarius*	5.28 ± 0.08	5.05 ± 0.17	95.68
CML352	*Lactobacillus salivarius*	5.20 ± 0.13	4.89 ± 0.07	93.96
CML346	*Lactobacillus salivarius*	5.07 ± 0.02	4.73 ± 0.01	93.39
CML312	*Lactobacillus johnsonii*	5.36 ± 0.02	4.97 ± 0.05	92.79
CML344	*Lactobacillus saerimneri*	5.60 ± 0.14	5.11 ± 0.05	91.27
CML345	*Lactobacillus salivarius*	5.49 ± 0.32	4.98 ± 0.07	90.67
CML104	*Ligilactobacillus agilis*	4.51 ± 0.07	4.06 ± 0.06	89.98
CML341	*Ligilactobacillus agilis*	4.51 ± 0.06	5.25 ± 0.10	86.89
CML333	*Lactobacillus johnsonii*	6.04 ± 0.15	3.59 ± 0.06	85.78
CML398	*Lactobacillus saerimneri*	4.19 ± 0.11	3.26 ± 0.14	82.32
CML323	*Ligilactobacillus agilis*	3.96 ± 0.06	4.35 ± 0.07	81.97
CML348	*Lactobacillus salivarius*	5.31 ± 0.03	4.18 ± 0.02	80.51

**Table 5 microorganisms-10-00726-t005:** Survival rate of different LAB strains under different bile salt concentrations.

Strain Number	Species	Treatments	
		0.3% Bile Salt Survival Rate (%)	0.5% Bile Salt Survival Rate (%)
CML319	*Lactobacillus johnsonii*	88.29	31.91
CML313	*Ligilactobacillus agilis*	42.70	41.25
CML352	*Lactobacillus salivarius*	64.24	47.08
CML350	*Lactobacillus salivarius*	30.81	27.79
CML346	*Lactobacillus salivarius*	66.64	39.83
CML398	*Lactobacillus saerimneri*	56.45	21.34
CML344	*Lactobacillus saerimneri*	52.85	45.67
CML104	*Ligilactobacillus agilis*	47.88	32.36

**Table 6 microorganisms-10-00726-t006:** Auto and co-aggregation ability and Hydrophobicity of lactic acid bacteria strains.

Strain Number	Species	Auto-Aggregation (%)	Co-Aggregation (%)			Hydrophobicity
			*Escherichia coli*	*Salmonella* *typhimurium*	*Clostridium* *perfringens*	
CML319	*Lactobacillus johnsonii*	84.14	48.49	56.21 ^abc^	49.05 ^b^	41.13 ^ab^
CML344	*Ligilactobacillus* *saerimneri*	81.16	48.34	49.33 ^bc^	48.26 ^b^	49.08 ^ab^
CML352	*Ligilactobacillus* *salivarius*	84.70	47.12	47.92 ^c^	46.54 ^b^	57.66 ^a^
CML398	*Ligilactobacillus* *saerimneri*	80.40	47.36	54.02 ^abc^	45.51 ^b^	41.66 ^ab^
CML104	*Ligilactobacillus* *agilis*	78.55	48.44	57.95 ^ab^	56.38 ^a^	21.21 ^b^
CML350	*Ligilactobacillus* *salivarius*	79.89	48.42	61.72 ^a^	48.57 ^b^	50.37 ^ab^

^a–c^ Within a row, means without a common superscript differ significantly (*p* < 0.05).

**Table 7 microorganisms-10-00726-t007:** Antimicrobial activity and hemolysis of selected lactic acid bacteria strains.

Strain	Species	Inhibition Zone			Hemolysis
		*Escherichia coli*	*Salmonella typhimurium*	*Clostridium perfringens*	
CML319	*Lactobacillus johnsonii*	−	−	++	α
CML350	*Ligilactobacillus salivarius*	+	++	+++	α
CML352	*Ligilactobacillus salivarius*	++	+	+++	α
CML344	*Ligilactobacillus saerimneri*	++	++	+++	γ
CML104	*Ligilactobacillus agilis*	+	+	++	α
CML398	*Ligilactobacillus saerimneri*	+	+	+++	γ
Control	MRS	−	−	−	
	AMP	+++	+++	+++	

Antimicrobial activity: +, <10 mm of inhibition; ++, between 10 and 15 mm of inhibition; +++, 15 mm of inhibition and above; −, no inhibition.

**Table 8 microorganisms-10-00726-t008:** Effects of dietary supplementation with *L. salivarius* on laying performance ^1^.

Items	Treatments		SEM ^2^	*p*-Value
	Control	CML352		
Egg production, %				
Weeks 56–59	90.44	90.18	0.079	0.891
Weeks 60–63	89.69	87.72	0.729	0.389
Weeks 64–67	84.95	82.86	0.896	0.345
Weeks 56–67	88.35	86.92	0.579	0.299
Average egg weight, g				
Weeks 56–59	62.46 ^a^	63.16 ^ab^	0.433	0.090
Weeks 60–63	63.27 ^a^	64.20 ^ab^	0.075	0.084
Weeks 64–67	63.56	63.86	0.016	0.632
Weeks 56–67	63.10 ^a^	63.74 ^b^	0.720	0.022
Average daily feed intake, g/hen per day				
Weeks 56–59	117.71	117.50	0.596	0.752
Weeks 60–63	112.35	112.38	0.291	0.975
Weeks 64–67	104.12 ^a^	98.33 ^b^	0.416	0.000
Weeks 56–67	111.40	109.41	0.518	0.719
Feed conversion ratio, g/g				
Weeks 56–59	2.08	2.06	0.062	0.669
Weeks 60–63	1.99	2.00	0.868	0.734
Weeks 64–67	1.94	1.87	0.836	0.241
Weeks 56–67	2.00	1.98	0.691	0.789

^1^*n* = 8 replicates per treatment. ^2^ SEM, standard error of the mean. ^ab^ Values within a row with no common superscripts differ significantly (*p* < 0.05).

**Table 9 microorganisms-10-00726-t009:** Effects of dietary supplementation with *L. salivarius* on weight and abdominal fat deposition of laying hens.

Items	Treatments		*p*-Value
	Control	CML352	
The average weight/kg	2.08 ± 0.01	2.00 ± 0.04	0.070
Abdominal fat proportion/%	5.19 ± 0.41	3.29 ± 0.69	0.037

**Table 10 microorganisms-10-00726-t010:** Effects of dietary supplementation with *L. salivarius* on egg quality of laying hens.

Items	Treatments		*p*-Value
	Control	CML352	
eggshell strength/Pa	3.43 ± 0.10	3.68 ± 0.08	0.042
Haugh unit	78.84 ± 1.68	82.08 ± 0.88	0.090
yolk proportion/%	0.27 ± 0.00	0.28 ± 0.00	0.605
eggshell thickness/mm	0.32 ± 0.01	0.34 ± 0.00	0.023
albumen height/mm	6.67 ± 0.21	6.94 ± 0.13	0.286

## Data Availability

The new nucleic acid sequences have been deposited at GenBank under the accession SRP361917. The version described in this paper is version SRP361917.

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
