# Peer review of "Lactobacillus salivarius CML352 Isolated from Chinese Local Breed Chicken Modulates the Gut Microbiota and Improves Intestinal Health and Egg Quality in Late-Phase Laying Hens"

_microorganisms, 2022, doi:10.3390/microorganisms10040726_

Round 1

Reviewer 1 Report

In the article entitled: "Lactobacillus salivarius CML352 isolated from Chinese local breed chicken modulates the gut microbiota and improves intestinal health and egg quality in late-phase laying hens" authors provided a well-designed study regarding the whole process of obtaining and testing gut microbiota of lying hens in order to find probiotic strains.  Basing on the study authors have selected LAB strain CML352, and evaluated its influence in vivo.

The study provided by authors is interesting, introduction is sufficient, methods are sound, results are clearly described.

I recommend publishing this study after applying some minor changes:

  1. Please define in which comparison authors used one way Anova, two-way Anova or Student’s t-test as declared?
  2. Author declared that results are presented with SD (section 2.8), meanwhile in the figure 5 we may found results with SEM (please reveal that)
  3. Reveal the LAB abbreviation when used for the first time
  4. Please provide the PCR conditions or adequate references in section 2.7.6

I have no further comments.

Reviewer 2 Report

Comments for the authors

  • L86: provide details about the ethics approval of the study
  • L92-101: provide appropriate references
  • L142-148: provide appropriate references
  • L187: specify in table the following as footnotes: Corn [7.8%], Soybean meal [43%], Corn gluten meal [60%], L-lysine-HCl [78%], Choline chloride50%, Vitamin premix, Ethoxyquin MAX, Phytase10000
  • L206-216: provide appropriate references

Reviewer 3 Report

 Peer review report:

  1. Original Submission

1.1. Recommendation: Minor Revision

  1. Comments to authors

Overview and general recommendation:

Overall, the study is well designed and well performed. The resulats are clearly presented. The discussion was clear. The manuscript is well structured. Although the flaws within the manuscript, I suggest its publication in case of minor revision.
Some indications for minor revisions are given below.

Line 39 or Line 50: Add the references Zommiti et al. (2019) and Zommiti et al. (2021). Two pertinent references firmly associated to the field.

Line 61: "...might dependent..". It should be corrected as "...might depend...".

Line 80: "Based on these, ...". Try to correct it as follows: "Based on these findings, ...". It is more suitable for the meaning.

Line 115: Try to eliminate the "%" after 100. It is mentioned at the beginning as a Survival rate (%).

Line 133: "...was evaluated..". Try to change it as follows: "...was performed according...". It is more adequate.

Line 143: Try to put the term "Briefly" in order to describe the experiment. So, it will be as follows: "...by Oxford Cup method. Briefly, 100 μL solution of...".

Line 188: "XOS": ???

Line 202: "..the eggshells were thickness...". Change it to make a sens to the sentence.

Line 249 to Line 251: Reformulate the sentence to make it clearer.

Line 255 to Line 258: Try to apply the new taxonomy of Lactobacillus spp. genus.

Line 277: "...their ability to tolerant pH..". Change it as follows: "their ability to tolerate pH...". 

Line 290: "E.coli, Escherichia coli." Apply the italic mode.

Line 300: Within the table 7: Clostridium Perfringens. "perfringens" should not be written with capital letter.

Line 304 to line 314: You did not put the Figure 1B ! Try to put it in the line 309 as follows: "and 63 RNA genes were predicted (Fig. 1B).

Line 335: "...of strain CML352...". Change it as follows: "of the CML352 strain..". It is more suitable.

Line 349: Reformulate it.

Line 360: "...indicated the microbiota..". Change it as follows: "...indicated that the microbiota...". It is more adequate.

Line 366:  "OTUs" ???

Line 417: There is no Figure 5. It should be changed to Figure 4.

Line 453 - Line 472: "in vitro" should be written in italics.

Line 454 to line 456: Reformulate it.

Line 463: Try give more references about Acid and bile salt tolerance with Lactobacillus species.

Line 473: Try to enrich this discussion section with other references.

Line 478: The same remark, try to add recent references to develop the discussion.

Line 535: E. coli. Put it in italics.

Line 558: "...other Lactobacillus strain...". Correct it as follows: "...other Lactobacillus strains...". It is more adequate.

Line 569: " L. salivarius". Put it in italics.

Line 582: "as feed addictive" ? You mean feed additive ?

Line 584: Try to give more references not only about L. salivarius but about other Lb. strains and their role in improving such parameters in terms of egg or laying hens, etc.

Line 588:  "...this candidate probiotic...". Inverse it as follows: "the probiotic candidate...".

Determination of the Ability of Lb. salivarius to Adhere to Caco-2 Cells: Have you ever thought to study in vitro the intestinal cell barrier integrity through measuring Transepithelial electrical resistance (TEER) with Caco-2 cells ? It seems a pertinent tool in order to study the barrier integrity.

Antibiotic Susceptibility and Minimal Inhibitory Concentration Determination: there is no data or mention of this imortant parameter specially in terms of probiotic safety.

β-Galactosidase Activity: a simple test that gives you idea if the Lb. salivarius strain is able to produce this important enzyme !

Enzymatic Activities of Lb. salivarius: a simple kit of API-ZYM can give a whole idea about the arsenal of enzymes produced by this strain. So, you emphasize the probiotic potential of the CML352 strain ! 

You have mentioned that there was antimicrobial activity, but you have not associated this microbial antagonism to any factor ! Was it a BLIS, a bacteriocin or another factor? Have you ever thought to search for genes of bacteriocins?

The yolk cholesterol is an important factor determining the quality of eggs. Why you did not measure it?  

Try to enrich the discussion section with recent references in terms of each parameter.

Try to add few more lines in the conclusion as perspectives and future use of this strain.

Check all the text in order to put commas and punctuation in right places to give the meaning to the sentences and ideas.

Check all the text for italic mode.

English language and style are fine. But a check is required

Round 2

Reviewer 1 Report

I've got no further comments. Thank you.

Author Response

We would like to thank you for your careful reading, helpful comments, and constructive suggestions, which has significantly improved the presentation of our manuscript.

Reviewer 2 Report

I accept the response to my comments

Author Response

(The authors gave the same response as above.)

Reviewer 3 Report

We thank you for all the answers and the included modifications. However, we want to clear some points:

Point 32: As you said in the author's reply: " the tolerance of this strain to acid and bile acid salt has higher priority...". No, it is not true, tolerance to acidity and to bile salts is an essential criterion, but there are a lot of other parameters which are most crucial in terms of probiotic criteria of LAB.

Regarding the survival of the strain in vivo conditions, you should perform tolerance to lyzozyme test in addition to acides and bile salts.

In terms of safety for animal and human consumption, Antibiotic Susceptibility represents one of the most salient criterion of safety assessment of any potential probiotic strain. Studies have revealed diverse strains with astonishing probiotic potential but, they were classified because of high resistance to antibiotics.

P.S: NOT ALL the LAB strains are “generally regarded as safe (GRAS). a wide range of LAB don not have this status.

Point 33: β-Galactosidase and its activity are so important notably for humans. Try in the next paper to investigate the production of this enzyme by the CML352 strain. (See the references Zommiti et al. 2016, Zommiti et al. 2017 and the isolation of Lb. curvatus DN317 strain from Saudi
chicken ceca, they are highly connected to your study). I recommennd to include these two pertinent references within your investigation.

Point 35: Antimicrobial activity of the strain CML352 is so important especially in terms of fighting sturdy pathogens in laying hens, which are so vulnerable. Try to investigate in the genome sequence for potential bacteriocin genes.

  • Just try to include the references Zommiti et al. 2016, and Zommiti et al. 2017 (highly related to the field of your research), 

Thank you again for all the clear answers.

Recommendation: Minor Revision 
